# Compositional Generalization via Neural-Symbolic Stack Machines

**Xinyun Chen**
UC Berkeley
xinyun.chen@berkeley.edu

**Chen Liang, Adams Wei Yu**
Google Brain
{crazydonkey,adamsyuwei}@google.com

**Dawn Song**
UC Berkeley
dawnsong@cs.berkeley.edu

**Denny Zhou**
Google Brain
dennyzhou@google.com

## Abstract

Despite achieving tremendous success, existing deep learning models have exposed limitations in compositional generalization, the capability to learn compositional rules and apply them to unseen cases in a systematic manner. To tackle this issue, we propose the Neural-Symbolic Stack Machine (NeSS). It contains a neural network to generate traces, which are then executed by a symbolic stack machine enhanced with sequence manipulation operations. NeSS combines the expressive power of neural sequence models with the recursion supported by the symbolic stack machine. Without training supervision on execution traces, NeSS achieves 100% generalization performance in four domains: the SCAN benchmark of language-driven navigation tasks, the task of few-shot learning of compositional instructions, the compositional machine translation benchmark, and context-free grammar parsing tasks.

## 1 Introduction

Humans have an exceptional capability of compositional reasoning. Given a set of basic components and a few demonstrations of their combinations, a person could effectively capture the underlying compositional rules, and generalize the knowledge to novel combinations [9, 36, 29, 28]. In contrast, deep neural networks, including the state-of-the-art models for natural language understanding, typically lack such generalization abilities [26, 24, 32], although they have demonstrated impressive performance on various applications.

To evaluate the compositional generalization, [26] proposes the SCAN benchmark for natural language to action sequence generation. When SCAN is randomly split into training and testing sets, neural sequence models [3, 19] can achieve perfect generalization. However, when SCAN is split such that the testing set contains unseen combinations of components in the training set, the test accuracies of these models drop dramatically, though the training accuracies are still nearly 100%. Some techniques have been proposed to improve the performance on SCAN, but they either still fail to generalize on some splits [42, 27, 30, 16, 1], or are specialized for SCAN-like grammar learning [38].

In this paper, we propose the **Ne**ural-**S**ymbolic **S**tack machine (NeSS), which integrates a symbolic stack machine into a sequence-to-sequence generation framework, and learns a neural network as the controller to operate the machine. NeSS preserves the capacity of existing neural models for sequence generation; meanwhile, the symbolic stack machine supports recursion [6, 8], so it can break down the entire sequence into components, process them separately and then combine the results, encouraging the model to learn the primitives and composition rules. In addition, we propose the notion of *operational equivalence*, which formalizes the intuition that semantically similar sentences

Table 1: Instruction semantics of our stack machine. See Figure 1 for the sample usage.

| Operator | Arguments | Description |
|---|---|---|
| SHIFT | — | Pull one token from the input stream to append to the end of the stack top. |
| REDUCE | $[t_1, t_2, ..., t_l]$ | Reduce the stack's top to a sequence $[t_1, t_2, ..., t_l]$ in the target language. |
| PUSH | — | Push a new frame to the stack top. |
| POP | — | Pop the stack top and append the popped data to the new stack top. |
| CONCAT_M | $[i_1, i_2, ..., i_l]$ | Concatenate the items from the stack top and the memory with indices $i_1, i_2, ..., i_l$, then put the concatenated sequence in the memory. |
| CONCAT_S | $[i_1, i_2, ..., i_l]$ | Concatenate the items from the stack top and the memory with indices $i_1, i_2, ..., i_l$, then put the concatenated sequence in the stack top. |
| FINAL | — | Terminate the execution, and return the stack top as the output. |

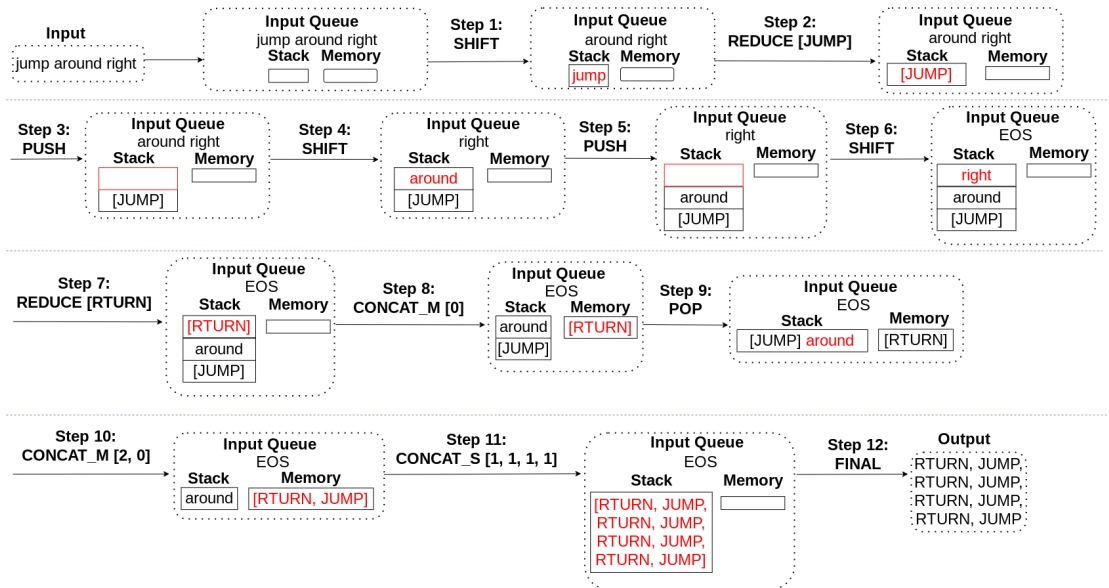

Figure 1: An illustrative example of how to use the stack machine for SCAN benchmark. A more complex example can be found in the supplementary material.

often imply similar operations executed by the symbolic stack machine. It enables NeSS to categorize components based on their semantic similarities, which further improves the generalization. To train our model without the ground truth execution traces, we design a curriculum learning scheme, which enables the model to find correct execution traces for long training samples.

We evaluate NeSS on four benchmarks that require compositional generalization: (1) the SCAN benchmark discussed above; (2) the task of few-shot learning of compositional instructions [28]; (3) the compositional machine translation task [26]; and (4) the context-free grammar parsing tasks [8]. NeSS achieves $100\%$ generalization performance on all these benchmarks.

## 2 Neural-Symbolic Stack Machine (NeSS)

In this section, we demonstrate NeSS, which includes a symbolic stack machine enhanced with sequence manipulation operations, and a neural network as the machine controller that produces a trace to be executed by the machine. We present our stack machine in Section 2.1, describe the model architecture of our machine controller in Section 2.2, and discuss the expressiveness and generalization power of NeSS in Section 2.3.

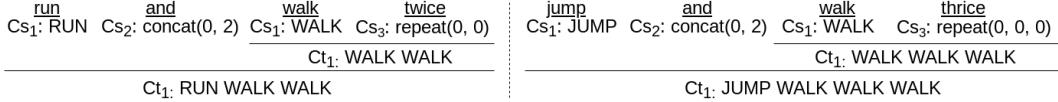

Figure 2: An illustration of component categorization, where $Cs_i$ and $Ct_i$ denote the $i$-th category of source and target languages respectively.

## 2.1 Enhanced Stack Machine for Sequence Manipulation

We design a stack machine that supports recursion, a key component to achieving compositional generalization. In particular, this machine supports general-purpose sequence-to-sequence tasks, where an input sequence in a source language is mapped into an output sequence in a target language. We present an overview of the machine operations in Table 1. Specifically, SHIFT is used for reading the input sequence, PUSH and POP are standard stack operations, REDUCE is used for output sequence generation, CONCAT_M and CONCAT_S concatenate the generated sequences to form a longer one, and the FINAL operation terminates the machine operation and produces the output. We provide an illustrative example in Figure 1, and defer more descriptions to the supplementary.

### 2.1.1 Operational Equivalence

Inspired by Combinatory Categorial Grammar (CCG) [43], we use component categorization as an ingredient for compositional generalization. As shown in Figure 2, from the perspective of categorical grammars, categories for source language may be considered as the primitive types of the lexicon, while predicting categories for the target language may be considered as the type inference. By mapping "jump" and "run" into the same category, we can easily infer the meaning of "run and walk" after learning to "jump and walk". Meanwhile, mapping "twice" and "thrice" into the same category indicates the similarity of their combinatorial rules, e.g., both of them should be processed before parsing the word "and". From the perspective of parsing, categorical information is encoded in non-terminals of the (latent) parse tree, which provides higher-level abstraction of the terminal tokens' semantic meaning. However, annotations of tree structures are typically unavailable or expensive to obtain. Faced with this challenge similar to unsupervised parsing and grammar induction [2, 11, 5], we leverage the similarities between the execution traces to induce the latent categorical information. This intuition is formalized as *operational equivalence* below.

**Operational Equivalence (OE).** Let $\mathcal{L}_s$, $\mathcal{L}_t$ be the source and target languages, $\pi$ be a one-to-one mapping from $\mathcal{L}_s$ to $\mathcal{L}_t$; $Op_\pi(s)$ be the operator to perform the mapping $\pi$, given the current machine status $s$; $\mathcal{S}$ be the set of valid machine statuses; $s' = R(s, s_i, s_i')$ means replacing the occurrences of $s_i$ in $s$ with $s_i'$. Components $s_i$ and $s_i'$ are operationally equivalent if and only if $\forall s \in \mathcal{S}, s' = R(s, s_i, s_i') \in \mathcal{S}$ and $Op_\pi(s) = Op_\pi(s')$.

In Figure 3, we present some examples of operational equivalence captured by the execution traces. We observe that, when two sequences only differ in the arguments of REDUCE, their corresponding tokens could be mapped to the same category, which is the main focus of most prior work on compositional generalization [16, 30, 27]. For example, [16] proposes the notation of *local equivariance* to capture such information. On the other hand, by grouping sequences only differing in CONCAT_M and CONCAT_S arguments, we also allow the model to capture structural equivalence, as shown in Figure 3b, which is the key to enabling generalization beyond primitive substitutions.

## 2.2 Neural Controller

With the incorporation of a symbolic machine into our sequence-to-sequence generation framework, NeSS does not directly generate the entire output sequence. Instead, the neural network in NeSS acts as a controller to operate the machine. The machine runs the execution trace generated by the neural network to produce the output sequence. Meanwhile, the design of our machine allows the neural controller to make predictions based on the local context of the input, which is a key factor to achieving compositional generalization. We provide an overview of the neural controller in Figure 4, describe the key components below, and defer more details to the supplementary.

**Machine status encoder.** A key property of NeSS is that it enables the neural controller to focus on the local context that is relevant to the prediction, thanks to the recursion supported by the stack

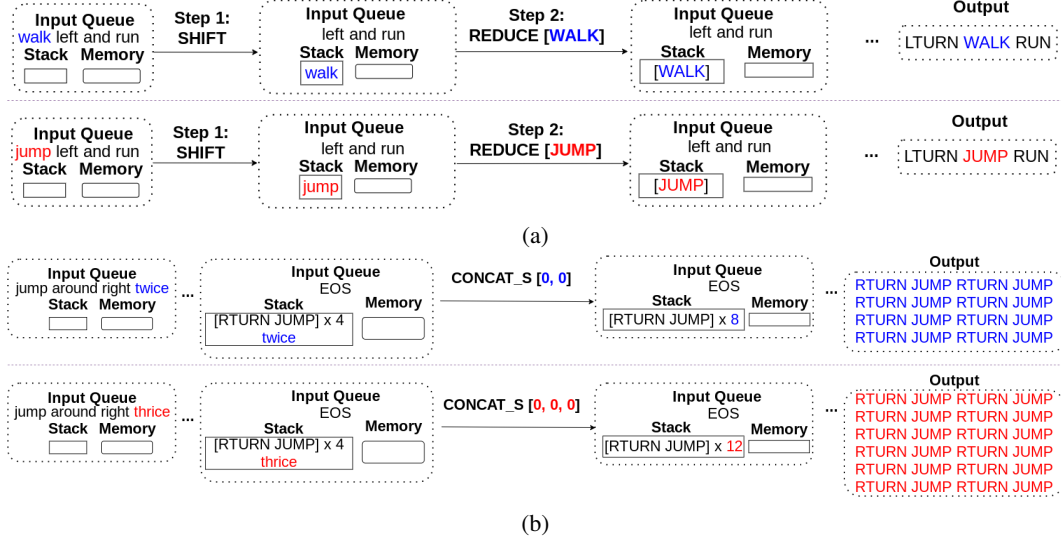

(a)

(b)

Figure 3: An illustration of the operational equivalence captured by the execution traces on SCAN benchmark. (a) With primitive replacement, e.g., changing "walk" into "jump", the operator trace remains the same, while the REDUCE arguments differ, thus "walk" and "jump" can be grouped into the same category. Such equivalence is also characterized by local equivariance defined in [16]. (b) By changing "twice" into "thrice", the operator trace remains the same, while the CONCAT_M and CONCAT_S arguments could differ, thus "twice" and "thrice" are in the same category. Such equivalence is crucial in achieving length generalization on SCAN, which is not characterized by primitive equivariance studied in prior work [16, 30, 27]

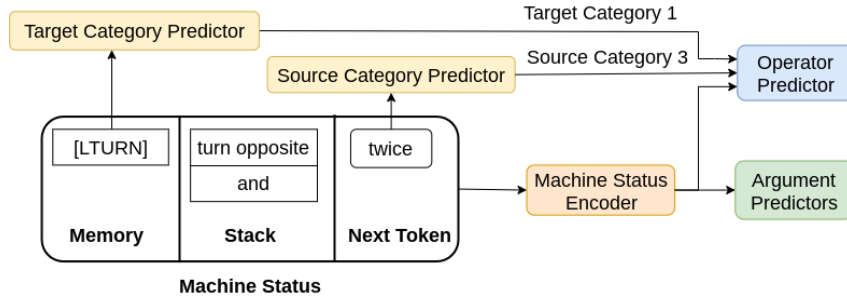

Figure 4: An overview of the neural architecture for the machine controller. A more detailed illustration is included in the supplementary material.

machine. Specifically, the input to the neural controller consists of three parts: (1) the next token in the input queue $tok$, e.g., the token "around" before executing step 4 in Figure 1; (2) the top 2 frames of the stack; and (3) the memory. Note that including the second stack frame from the top is necessary for determining the association rules of tokens, as discussed in [8]. Take arithmetic expression parsing as an example, when the top stack frame includes a variable "y" and the next input token is "*", we continue processing this token when the previous stack frame includes a "+", while we need a POP operation when the previous stack frame includes "*". We use 4 bi-directional LSTMs as the machine status encoders to encode the input sequence, the top 2 stack frames and the memory respectively. Then we denote the 4 computed embedding vectors as $e_{tok}$, $e_{cur}$, $e_{pre}$ and $e_M$.

**Operator predictor.** The operator predictor is a multi-layer fully connected network with a $|Op|$-dimensional softmax output layer, where $|Op| \models 7$ is the number of operators supported by the machine, as listed in Table 1. Its input is the concatenation of $e_{tok}$, $e_{cur}$, $e_{pre}$ and $e_M$.

**Argument predictors.** We include three neural modules for argument prediction, which are used for REDUCE, CONCAT_M and CONCAT_S respectively. We design the REDUCE argument predictor as a standard LSTM-based sequence-to-sequence model with attention, which generates a token sequence in the target vocabulary as the arguments. For both CONCAT_M and CONCAT_S argument predictors,

we utilize the pointer network architecture [46] to select indices from an element list as arguments, where the list contains the elements from the top stack frame and the memory.

**Latent category predictors.** We introduce two latent category predictors for source and target languages. The source category predictor, denoted as $p_{\text{sc}}(e_{tok})$, computes an embedding vector $ec_{tok}$ to indicate the categorical information given the input $e_{tok}$. Similarly, we denote the target category predictor as $p_{\text{tc}}(e_s)$, where the input $e_s$ is the embedding of the token sequence $s$ in the target language. For the input to the operator predictor, we replace $e_{tok}$ with $ec_{tok}$ as the representation of the next input token $tok$, which encourages the neural controller to predict the same operator for tokens of the same category. Similarly, the categorical predictions for the target language are used for subsequent instruction predictions.

### 2.3 Discussion

In the following, we discuss the expressiveness and generalization power of NeSS. In particular, NeSS preserves the same expressive power as sequence-to-sequence models, while our neural-symbolic design enhances its compositional generalization capability.

**Expressive power.** When we impose no constraints to regularize the machine, e.g., we do not restrict the argument length for REDUCE instruction, there is a *degenerate* execution trace that is valid for every input-output example. Specifically, this trace keeps running the SHIFT instruction until an [EOS] is observed as the next input token, then executes a REDUCE instruction with the entire output sequence as its argument. In this way, NeSS preserves the capacity of existing sequence-to-sequence models [3, 48, 19, 44, 13]. To leverage the recursion property of the machine, we could set a length limit for REDUCE arguments, so that the neural model mainly calls REDUCE instruction to generate phrases in the target language, and utilizes other instructions to combine the generated primitives to form a longer sequence. We call such compositional traces as *non-degenerate* traces hereafter.

**Generalization.** Some recent work proposes neural-symbolic architectures to achieve length generalization for program induction [6, 8, 49, 39]. The core idea is to incorporate a stack into the symbolic machine, so that the neural network model could restrict its attention to only part of the input important for the current decision. This recursion design principle is also crucial in achieving compositional generalization in our case. Meanwhile, capturing the operational equivalence enables NeSS to simultaneously obtain generalization capability and expressive power.

## 3 Training

As discussed in Section 2.3, without the ground truth execution traces as training supervision, the model may exploit the REDUCE argument generator and predict degenerate traces without compositionality. To avoid this degenerate solution, we apply a curriculum learning scheme. At a high level, the model first performs a trace search for each sample in the lesson, and then categorizes components based on the operational equivalence. We discuss the key sub-routines below. The full training procedure can be found in the supplementary material.

**Curriculum learning.** We sort the training samples in the increasing order of their input and output lengths to construct the curriculum. Before training on the first lesson, the neural model is randomly initialized. Afterwards, for each new lesson, the model is initialized with the parameters learned from previous lessons. To obtain training supervision, for each input sequence within the current lesson, we search for an execution trace that leads to the ground truth output, based on the probability distribution predicted by the neural model, and we prioritize the search for non-degenerate execution traces. If the model could not find any non-degenerate execution trace within the search budget, a degenerate execution trace is used for training. The model proceeds to the new lesson when no more non-degenerate execution traces can be found.

**Learning to categorize.** To provide training supervision for latent category predictors, we leverage the operational equivalence defined in Section 2.1.1. Specifically, after the trace search, we collect the non-degenerate operator traces, and compare among different samples within a training batch. If two samples share the same operator trace, we first assign their target sequences with the same target category for training. Note that their input sequences have the same length, because the same SHIFT

instructions are executed. Therefore, we enumerate each index within the input length, pair up the tokens with the same index, and assign them with the same source category for training.

## 4 Experiments

We evaluate NeSS in four domains: (1) the SCAN splits that require compositional generalization [26]; (2) the task of few-shot learning of compositional instructions proposed in [28]; (3) the compositional machine translation benchmark proposed in [26]; and (4) the context-free grammar parsing benchmarks proposed in [8]. We present the setup and key results below, and defer more experimental details to the supplementary material. Note that we perform greedy decoding to generate the execution traces during the inference time, without any search.

### 4.1 SCAN Benchmark

The SCAN benchmark has been widely used to evaluate the compositional generalization of neural networks, where the input sequence is a navigation command in English, and the output is the corresponding action sequence [26]. See Figure 1 for a sample usage of NeSS for the SCAN tasks.

**Evaluation setup.** Similar to prior work [27, 16, 38], we evaluate the following four settings. (1) **Length generalization**: the output sequences in the training set include at most 22 actions, while the output lengths in the test set are between 24 and 48. (2) **Template generalization for "around right"**: the phrase "around right" is held out from the training set; however, both "around" and "right" occurs in the training set separately. For example, the phrases "around left" and "opposite right" are included in the training set. (3) **Primitive generalization for "jump"**: all commands not including "jump" are included in the training set, but the primitive "jump" only appears as a single command in the training set. The test set includes commands combining "jump" with other primitives and templates, such as "jump twice" and "jump after walk". (4) **Simple split**: randomly split samples into training and test sets. In this case, no compositional generalization is required.

**Previous approaches.** We compare NeSS with two classes of existing approaches on SCAN benchmark. The first class of approaches propose different neural network architectures, without additional data to provide training supervision. Specifically, sequence-to-sequence models (seq2seq) [26] and convolutional neural networks (CNN) [12] are standard neural network architectures, Stack LSTM learns an LSTM to operate a differentiable stack [18], while the equivariant sequence-to-sequence model [16] incorporates convolution operations into the recurrent neural networks, so as to achieve local equivariance discussed in Section 2.1.1. On the other hand, the syntactic attention model [42] and primitive substitution [30] learn two attention maps for primitives and templates separately. The second class of approaches design different schemes to generate auxiliary training data. Specifically, GECA [1] performs data augmentation by replacing fragments of training samples with different fragments from other similar samples, while the meta sequence-to-sequence model [27] and the rule synthesizer model (synth) [38] are trained with samples drawn from a meta-grammar with the format close to the SCAN grammar.

Table 2: Learned categories on SCAN. The words in a pair of brackets belong to the same category. The categories contained in the three lines are respectively learned from input sequences of length 1, 2 and 3.

| {run, look, jump, look} |
| {left, right}, {twice, thrice}, {turn} |
| {and, after}, {around}, {opposite} |

**Results.** Table 3 summarizes our results on SCAN tasks. In the top block, we compare with models trained without additional data. Among these approaches, NeSS is the only one achieving $100\%$ test accuracies on tasks that require compositional generalization, and the performance is consistent among 5 independent runs. In particular, the best generalization accuracy on the length split is only around $20\%$ for the baseline models. Note that the stack LSTM does not achieve better results, demonstrating that without a symbolic stack machine that supports recursion and sequence manipulation, augmenting neural networks with a stack alone is not sufficient. Meanwhile, without category predictors, NeSS still achieves $100\%$ test accuracy in 2 runs, but the accuracy drops to around $20\%$ for other 3 runs. A main reason is that existing models could not generalize to the input template "around left/right thrice", when the training set only includes the template "around left/right twice". Although NeSS correctly learns the parsing rules for different words, without category predictors, NeSS still may not learn

that the parsing rule for "thrice" has the same priority as "twice". For example, in the test set, there is a new pattern "jump around right thrice". The correct translation is to parse "jump around right" first, then repeat the action sequence thrice, resulting in 24 actions. Without category prediction, NeSS could mistakenly parse "right thrice" first, concatenate the action sequences of "jump" and "right thrice", then repeat it for four times, resulting in 16 actions. Such a model could still achieve 100% training accuracy, because this pattern does not occur in the training set, but the test accuracy drops dramatically due to the wrong order for applying rules. Therefore, to achieve generalization, besides the parsing rules for each individual word, the model also needs to understand the order of applying different rules, which is not captured by the primitive equivalence [27, 30, 42] or local equivariance [16] studied in prior work. On the other hand, as shown in Table 2, the operational equivalence defined in Section 2.1.1 enables the model to learn the priorities of different parsing rules, e.g., categorizing "twice" and "thrice" together, which is crucial in achieving length generalization.

Next, we compare NeSS with models trained with additional data. In particular, the meta sequence-to-sequence model is trained with different permutations of primitive assignment, i.e., different one-to-one mapping of {run, look, jump, walk} to {RUN, LOOK, JUMP, WALK}, denoted as "(perm)". We consider two evaluation modes of the rule synthesizer model (synth) [38], where the first variant performs greedy decoding, denoted as "(no search)"; the second one performs a search process, where the model samples candidate grammars, and returns the one that matches the training samples. We observe that even with additional training supervision, Synth (with search) is the only baseline approach that is able to achieve 100% generalization on all these SCAN splits.

Although both Synth and NeSS achieve perfect generalization, there are key differences we would like to highlight. First, the meta-grammar designed in Synth restricts the search space to only include grammars with a similar format to the SCAN grammar [38]. For example, each grammar has between 4 and 9 primitive rules that map a single word to a single primitive (e.g., run $\rightarrow$ RUN), and 3 to 7 higher order rules that encode variable transformations given by a single word (e.g., x1 and x2 $\rightarrow$ [x1] [x2]). Therefore, Synth cannot be applied to other two benchmarks in our evaluation. Unlike Synth, NeSS does not make such assumptions about the number of rules nor their formats. Also, NeSS does not perform any search during the inference, while Synth requires a search procedure to ensure that the synthesized grammar satisfies the training samples.

## 4.2 Few-shot Learning of Compositional Instructions

Next, we evaluate on the few-shot learning benchmark proposed in [28], where the model learns to produce abstract outputs (i.e., colored circles) from pseudowords (e.g., "dax"). Compared to the SCAN benchmark, the grammar of this task is simpler, with 4 primitive rules and 3 compositional rules. However, while the SCAN training set includes over 10K examples, there are only 14 training samples in this benchmark, thus models need to learn the grammar from very few demonstrations. In [28], they demonstrate that humans are generally good at such few-shot learning tasks due to their inductive biases, while existing machine learning models struggle to obtain this capability.

**Results.** We present the results in Table 4, where we compare with the standard sequence-to-sequence model, the primitive substitution approach discussed in Section 4.1, and the human performance evaluated in [28]. We didn't compare with the meta sequence-to-sequence model and the rule synthesizer model discussed in Section 4.1, because they require meta learning with additional training samples. Despite that the number of training samples is very small, NeSS achieves 100% test accuracy in 5 independent runs, demonstrating the benefit of integrating the symbolic stack machine to capture the grammar rules.

## 4.3 Compositional Machine Translation

Then we evaluate on the compositional machine translation benchmark proposed in [26]. Specifically, the training set includes 11,000 English-French sentence pairs, where the English sentences begin with phrases such as "I am", "you are" and "he is", and 1,000 of the samples are repetitions of "I am daxy" and its French translation, which is the only sentence that introduces the pseudoword "daxy" in the training set. The test set includes different combinations of the token "daxy" and other phrases, e.g., "you are daxy", which do not appear in the training set. Compared to the SCAN benchmark, the translation rules in this task are more complex and ambiguous, which makes it challenging to be fully explained with a rigorous rule set.

Table 3: Test accuracy on SCAN splits. All models in the top block are trained without additional data. In the bottom, GECA is trained with data augmentation, while Meta Seq2seq (perm) and both variants of Synth are trained with samples drawn from a meta-grammar, with a format close to the SCAN grammar. In particular, Synth (with search) performs a search procedure to sample candidate grammars, and returns the one that matches the training samples; instead, other models always return the prediction with the highest decoding probability.

| Approach | Length | Around right | Jump | Simple |
|---|---|---|---|---|
| **NeSS (ours)** | **100.0** | **100.0** | **100.0** | **100.0** |
| Seq2seq [26] | 13.8 | – | 0.08 | 99.8 |
| CNN [12] | 0.0 | 56.7 | 69.2 | 100.0 |
| Stack LSTM [18] | 17.0 | 0.3 | 0.0 | 100.0 |
| Syntactic Attention [42] | 15.2 | 28.9 | 91.0 | – |
| Primitive Substitution [30] | 20.3 | 83.2 | 98.8 | 99.9 |
| Equivariant Seq2seq [16] | 15.9 | 92.0 | 99.1 | 100.0 |
| GECA [1] | – | 82 | 87 | – |
| Meta Seq2seq (perm) [27] | 16.64 | 98.71 | 99.95 | – |
| Synth (no search) [38] | 0.0 | 0.0 | 3.5 | 13.3 |
| Synth (with search) [38] | 100.0 | 100.0 | 100.0 | 100.0 |

Table 4: Accuracy on the few-shot learning task proposed in [28].

| Approach | Accuracy |
|---|---|
| **NeSS (ours)** | **100.0** |
| Seq2seq [30] | 2.5 |
| Primitive Substitution [30] | 76.0 |
| Human [28] | 84.3 |

Table 5: Accuracy on the compositional machine translation benchmark in [26], measured by semantic equivalence.

| Approach | Accuracy |
|---|---|
| **NeSS (ours)** | **100.0** |
| Seq2seq [26] | 12.5 |
| Primitive Substitution [30] | 100.0 |

Table 6: Results on the context-free grammar parsing benchmarks proposed in [8]. "Test-LEN" indicates the testset including inputs of length LEN.

| Test | **NeSS (ours)** | Neural Parser | Seq2seq | Seq2tree | Stack LSTM |
|---|---|---|---|---|---|
| Training | **100%** | 100% | 81.29% | 100% | 100% |
| Test-10 | **100%** | 100% | 0% | 0.8% | 0% |
| Test-5000 | **100%** | 100% | 0% | 0% | 0% |

**Results.** We present the results in Table 5, where we compare NeSS with the standard sequence-to-sequence model [26], and the primitive substitution approach discussed in Section 4.1. Note that instead of measuring the exact match accuracy, where the prediction is considered correct only when it is exactly the same as ground truth, we measure the semantic equivalence in Table 5. As discussed in [30], only one reference translation is provided for each sample in the test set, but there are 2 different French translations of "you are" that appear frequently in the training set, which are both valid translations. Therefore, if we measure the exact match accuracy, the accuracy of the Primitive Substitution approach is 62.5%, while NeSS achieves 100% in 2 runs, and 62.5% in 3 other runs. Although both NeSS and the Primitive Substitution approach achieves 100% generalization, by preserving the sequence generation capability of sequence-to-sequence models with the REDUCE argument generator, NeSS is the only approach that simultaneously enables length generalization for rule learning tasks and achieves 100% generalization on machine translation with more diverse rules, by learning the phrase alignment.

## 4.4 Context-free Grammar Parsing

Finally we evaluate NeSS on the context-free grammar parsing tasks in [8]. Following [8], we mainly consider the curriculum learning setting, where we train the model with their designed curriculum, which includes 100 to 150 samples enumerating all constructors in the grammar. NeSS parses the inputs by generating the serialized parse trees, as illustrated in the supplementary material. The average input length of samples in the curriculum is around 10. This benchmark is mainly designed to evaluate length generalization, where the test samples are much longer than training samples.

**Results.** We present the main results in Table 6, where we compare NeSS with the sequence-to-sequence model [47], sequence-to-tree model [15], LSTM augmented with a differentiable stack structure [18], and the neural parser [8]. All these models are trained on the curriculum of the While language designed in [8], and we defer the full evaluation results of more setups and baselines to the supplementary material, where we draw similar conclusions. Again, NeSS achieves $100\%$ accuracy in 5 independent runs. We notice that none of the models without incorporating a symbolic machine generalizes to test inputs that are $500 \times$ longer than training samples, suggesting the necessity of the neural-symbolic model design. Meanwhile, compared to the neural parser model, NeSS achieves the same capability of precisely learning the grammar production rules, while it supports more applications that are not supported by the neural parser model, as discussed in Section 2.3.

## 5 Related Work

There has been an increasing interest in studying the compositional generalization of deep neural networks for natural language understanding [26, 24, 4, 41]. A line of literature develops different techniques for the SCAN domain proposed in [26], including architectural design [42, 30, 16], training data augmentation [1], and meta learning [27, 38]. Note that we have already provided a more detailed discussion in Section 4.1. In particular, the rule synthesis approach in [38] also achieves $100\%$ generalization performance as NeSS. However, they design a meta-grammar space to generate training samples, which contains grammars with the format close to the SCAN grammar, and their model requires a search process to sample candidate grammars during the inference time. On the other hand, NeSS does not assume the knowledge of a restricted meta-grammar space as in [27, 38]. In addition, no search is needed for model evaluation, thus NeSS could be more efficient especially when the task requires more examples as the test-time input specification.

Some recent work also studies compositional generalization for other applications, including semantic parsing [24, 1], visual question answering [21, 4, 34, 45, 50, 20], image captioning [37], and other grounded language understanding domains [41]. In particular, a line of work proposes neural-symbolic approaches for visual question answering [34, 45, 50, 20], and the main goal is to achieve generalization to new composition of visual concepts, as well as scenes with more objects than training images. Compared to vision benchmarks measuring the compositional generalization, our tasks do not require visual understanding, but typically need much longer execution traces.

On the other hand, length generalization has been emphasized for program induction, where the learned model is supposed to generalize to longer test samples than the training ones [17, 51, 40, 6, 8]. A line of approaches learn a neural network augmented with a differentiable data structure or a differentiable machine [17, 51, 22, 23, 25, 18]. However, these approaches either can not achieve length generalization, or are only capable of solving simple tasks, as also shown in our evaluation. Another class of approaches incorporate a symbolic machine into the neural network [40, 6, 52, 8, 49, 39], which enables length generalization either with training supervision on execution traces [6], or well-designed curriculum learning schemes [8, 49, 39]. In particular, our neural-symbolic architecture is inspired by the neural parser introduced in [8], which designs a parsing machine based on classic SHIFT-REDUCE systems [10, 35, 31, 8]. By serializing the target parse tree, our machine fully covers the usages supported by the parsing machine. Meanwhile, the incorporation of a memory module and enhanced instructions for sequence generation enables NeSS to achieve generalization for not only algorithmic induction, but also natural language understanding domains. Some recent work also studies length generalization for other tasks, including relational reasoning [14], multi-task learning [7], and structure inference [33].

## 6 Conclusion

In this work, we presented NeSS, a differentiable neural network to operate a symbolic stack machine supporting general-purpose sequence-to-sequence generation, to achieve compositional generalization. To train NeSS without supervision on ground truth execution traces, we proposed the notation of operational equivalence, which captured the primitive and compositional rules via the similarity of execution traces. NeSS achieved $100\%$ generalization performance on four benchmarks ranging from natural language understanding to grammar parsing. For future work, we consider extending our techniques to other applications that require the understanding of compositional semantics, including grounded language understanding and code translation.

## Broader Impact

Our work points out a promising direction towards improving compositional generalization of deep neural networks, and has potential to be utilized for a broad range of applications. In practice, the neural-symbolic framework design enables people to impose constraints on the neural network predictions, which could improve their interpretability and reliability.

## Acknowledgments and Disclosure of Funding

This work was done while Xinyun Chen is a part-time student researcher at Google Brain.

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
