[Supplementary Material]

## A   Discussion of the Benchmark Selection for Evaluation

Given that NeSS achieves impressive results on synthetic natural language benchmarks in our evaluation, one question is whether it could also improve the performance on commonly used natural language datasets, e.g., large-scale machine translation benchmarks. However, note that most existing natural language benchmarks are not designed for evaluating the compositional generalization performance of models. Instead, the main challenge of those datasets is to handle the inherently ambiguous and potentially noisy natural language inputs. Specifically, their training and test sets are usually from the same distribution, and thus do not evaluate compositional generalization. As a result, we did not run experiments on these datasets. Instead, our evaluation focuses on standard sequence-to-sequence generation benchmarks used in previous works on compositional generalization. Such benchmarks are typically constructed with synthetic grammars, so that it is easier to change training and test distributions. We consider improving compositional generalization for more natural inputs as future work.

## B   More Details of the Stack Machine

We present the sample usage of our machine with a more complex example on SCAN benchmark in Figure 5. In the following, we provide a more detailed explanation of our `CONCAT_M` and `CONCAT_S` operations based on this example. Specifically, when executing the `CONCAT_M` operation at step 10, we first concatenate all items in the stack top and the memory as a list, i.e., [[JUMP], around, [RTURN]] in this case. Next, according to the argument [2, 0], the items with indices 2 and 0 are selected and concatenated, which results in the sequence [RTURN, JUMP]. Afterwards, this new sequence replaces the original content in the memory, and the selected item in the stack top, i.e., [JUMP], is removed from the stack top. On the other hand, the token "around" is kept in the stack top, because it is not selected for `CONCAT_M`. The argument selection for `CONCAT_S` is similar, except that this operation puts the generated sequence in the stack top.

In Figure 6, we present how our machine supports the `REDUCE` operation defined in the parsing machine of [8], which is designed for context-free grammar parsing.

## C   More Details of the Neural Controller Architecture

We present the neural controller architecture in Figure 7, and we describe the details below.

**Machine status encoder.**   We use a bi-directional LSTM $LSTM_{inp}$ to encode the token sequence in the input queue, and use the LSTM output for $tok$ as its vector representation $e_{tok}$. We use two separate bi-directional LSTMs $LSTM_{cur}$ and $LSTM_{pre}$ to encode the top 2 stack frames respectively. The LSTM output at the last timestep is used as the embedding of the 2 stack frames, denoted as $e_{cur}$ and $e_{pre}$. Similarly, another LSTM $LSTM_M$ is used to encode the memory, and we denote the embedding as $e_M$. Note that we always add an `[EOS]` token when computing the embedding for stack frames and memory, even if they are empty.

**REDUCE argument predictor.**   The `REDUCE` instruction takes the top stack frame as the input, and outputs a token sequence in the target vocabulary as the arguments, denoted as $p_{\text{REDUCE}}(arg|e_{cur})$. We design the `REDUCE` argument predictor as a standard LSTM-based sequence-to-sequence model with attention, and the argument generation process terminates when an `[EOS]` token is predicted. The output at the last timestep of the LSTM decoder is used as the embedding of the entire reduced sequence, and it replaces the embedding vectors originally in the top stack frame.

**CONCAT_M and CONCAT_S argument predictors.**   We design the same architecture for `CONCAT_M` and `CONCAT_S` argument predictors, but with different sets of model parameters, and we discuss the details for `CONCAT_M` as follows. Firstly, we use a bi-directional LSTM to compute an embedding vector for each element in the top 2 stack frames and the memory. Note that the stack frames and memory include tokens in the source vocabulary that are directly moved from the input queue using the `SHIFT` instruction, as well as sequences generated by previous `REDUCE`, `CONCAT_M` and `CONCAT_S` instructions that consist of tokens in the target vocabulary. To select the arguments for `CONCAT_M` and `CONCAT_S`, we only consider token sequences in the

Figure 5: A more complicated usage of the stack machine for SCAN benchmark.

Figure 6: An illustrative example of our stack machine for context-free grammar parsing. This example showcases the execution steps that are equivalent to a REDUCE operation defined in the parsing machine of [8]. CONCAT_M is used to select the children for the generated tree, REDUCE is used to generate the non-terminal, and CONCAT_S is used to construct the tree.

Figure 7: The neural architecture for the machine controller. The dotted arrows indicate the update of machine status representation after executing the corresponding instructions.

.

target vocabulary that are included in the top stack frame and the memory as the candidates. However, when computing the embedding vectors, we still include other elements in the top 2 stack frames and the memory, so as to encode the context information. We keep an additional embedding vector of the [EOS] token for argument prediction, which could be selected to terminate the argument generation process. We utilize a pointer network architecture [46] to select indices from the input element list as arguments, and the argument generation process terminates when it selects [EOS] token as the argument. The output at the last timestep of the pointer network is used as the embedding of the generated sequence, and it replaces the embedding vectors originally in the memory. We denote the two generators as $p_{\text{CONCAT\_M}}(arg|e_{cur}, e_{pre}, e_M)$ and $p_{\text{CONCAT\_S}}(arg|e_{cur}, e_{pre}, e_M)$.

**Latent category predictors.** Both source and target category predictors include a classification layer followed by an embedding matrix. Specifically, for the source category predictor, given $e_{tok}$ as the input, the classification layer is a $|C_s|$-dimensional softmax layer, which predicts a probability distribution of the category that the input word $tok$ belongs to. Let $c_{tok}$ be the category that $tok$ belongs to, another embedding matrix $E_c$ is used to compute an embedding vector $ec_{tok}$. Similarly, given an embedding vector of a token sequence $s$ in the target language, denoted as $e_s$, the classification layer of the target category predictor predicts a $|C_t|$-dimensional probability distribution indicating the category of the sequence $s$, then another embedding matrix is used to compute an embedding vector describing the categorical information of the token sequence. Note that when a SHIFT instruction is executed, we still put the embedding vector $e_{tok}$ to the stack top instead of its categorical embedding, since different tokens in the same category could be processed with different REDUCE arguments. For example, "left" should be reduced into "LTURN", while "right" should be reduced into "RTURN". On the other hand, the categorical predictions for the target language are used for subsequent predictions of both operators and arguments. We set the number of categories $|C_s|$ and $|C_t|$ for source and target languages as their vocabulary sizes, to support the degenerate mapping that considers each token as a separate category.

---
**Algorithm 1** Training algorithm for NeSS
---

**Input:** training lessons $L$, the $i$-th lesson $L_i = \{(x_{ij}, y_{ij})\}_{j=1}^{N_i}$, a model $p^\theta$
The extracted ruleset $R \leftarrow \emptyset$
The current training data $D \leftarrow \emptyset$
**for** $L_i \in L$ **do**
    $D \leftarrow D \cup L_i$
    **repeat**
        **for** each batch $B_j = \{(x_k, y_k)\}_{k=1}^{|B_j|} \in D$ **do**
            $T_j \leftarrow \text{TraceSearch}(p^\theta, B_j, R)$
            $Ops_j \leftarrow$ operator traces in $T_j$
            $args_j$ = `REDUCE`, `CONCAT_M`, `CONCAT_S` arguments in $T_j$
            // $OEs$, $OEt$: latent category supervision for the source and target languages.
            $OEs_j, OEt_j \leftarrow \text{OEExtraction}(B_j, T_j)$
            $Loss_{\text{op}} \leftarrow -\log p_{\text{op}}^\theta(Ops_j)$, $Loss_{args} \leftarrow -\log p_{\text{args}}^\theta(args_j)$
            $Loss_{\text{OE}} \leftarrow -(\log p_{sc}^\theta(OEs_j) + \log p_{tc}^\theta(OEt_j))$
            $Loss \leftarrow Loss_{\text{op}} + Loss_{\text{args}} + Loss_{\text{OE}}$
            Update $\theta$ to minimize $Loss$
        **end for**
    **until** No more non-degenerate execution traces are found with the search.
    $R \leftarrow \text{RuleExtraction}(p^\theta, L_i)$
**end for**

---

## D  More Details for Training

We outline the training algorithm in Algorithm 1. In the algorithm, we denote the prediction probability distribution of the operator predictor as $p_{\text{op}}$, and the argument prediction probability distribution as $p_{\text{args}}$.

**Rule extraction.** Our recursive machine design enables us to extract rules learned from previous lessons. For each execution step in a learned trace, we denote a tuple of (machine status, operator) as an extracted rule for operator prediction, where the machine status includes the contents of the top 2 stack frames, the memory, and the next token $tok$ in the input queue.

Similarly, we keep 3 rule sets for `REDUCE`, `CONCAT_M` and `CONCAT_S` argument prediction respectively, where the machine status includes the information used as the input to the corresponding predictors. For example, the ruleset for `REDUCE` argument prediction includes tuples of (stack top, argument). Therefore, after we extract the rules from NeSS trained on SCAN, its `REDUCE` ruleset should be as follows: {(run, [RUN]), (jump, [JUMP]), (look, [LOOK]), (walk, [WALK]), (left, [LTURN]), (right, [RTURN]), (turn left, [LTURN]), (turn right, [RTURN])}. The extracted ruleset for the few-shot learning and context free grammar parsing tasks also largely follow the pre-defined ground truth grammar. For the compositional machine translation benchmark, the main extracted `REDUCE` rules include: {(i am, [je suis]), (i am not, [je ne suis pas]), (you are, [tu es]), (you are not, [tu n es pas]), (he is, [il est]), (he is not, [il n est pas]), (she is, [elle est]), (she is not, [elle n est pas]), (we are, [nous sommes]), (we are not, [nous ne sommes pas]), (they are, [elles sont]), (they are not, [elles ne sont pas]), (very, [tres]), (daxy, [daxiste])}.

Note that we do not extract rules for degenerate execution traces, unless the length of the output sequence is 1, which suggests that the degenerate execution trace is the most appropriate one.

**Speed up the trace search with extracted rules.** To further speed up the trace search during the training process, we utilize the rules extracted from previous lessons, and prioritize their usage for the trace search in the current lesson. In Figure 8, we provide some examples of spurious traces without leveraging the rules extracted from previous lessons. For example, in the spurious trace for "walk after jump" shown in Figure 8a, "walk" and "jump" are wrongly reduced into "JUMP" and "WALK" respectively, and with the wrong `CONCAT_S` argument, the output sequence still matches the ground truth. Besides the wrong arguments, a spurious trace could also get the operators wrong, as shown in Figure 8b. In this spurious trace, a `REDUCE` operation is applied to the word "twice". Given that "jump", "walk" and "twice" already appear in previous lessons including shorter sentences, ideally, a

Figure 8: Sample spurious traces on SCAN benchmark, which could be pruned by rule extraction. The wrong predictions of operators and arguments are marked with red.

well-trained model is supposed to perfectly memorize them. However, since our trace search is a sampling process, such spurious traces are still possible, especially when the input sequences become long. With the rule extraction process for training, NeSS prioritizes traces where the operators and arguments do not conflict with the learned rules, e.g., those with the correct REDUCE arguments for "walk" and "jump", and the correct operations for "twice". Specifically, when NeSS encounters a machine status that is already included in the rule set extracted from previous lessons, NeSS directly applies the corresponding rule, and only searches for other operations when it cannot find any trace consistent with the extracted rule.

**Training for latent category predictors.** During the training process, when we encounter two instances that are considered as potentially operationally equivalent, we first feed one of the instances into the latent category predictor, and randomly sample a category index based on the probability distribution computed by the predictor. Afterwards, we set this category index to be the ground truth category for both the two instances. If the first occurrence of one instance is in an earlier lesson than another one, then we sample the category index based on the prediction probability distribution computed for this instance, otherwise we arbitrarily select one instance from them.

# E   Implementation Details

**Curriculum design.** For SCAN benchmark, we split the training set into 6 lessons. The first 4 lessons include samples with an input sequence length or an output sequence length of 1, 2, 3 and 4

respectively. The fifth lesson includes all samples with an input sequence length larger than 4, and a maximal output action sequence length of 8. The sixth lesson includes the rest of training samples.

For the compositional machine learning benchmark, the curriculum is designed with the increasing order of length of the English sentences, where the first lesson includes the shortest sentences with 3 words, e.g., "I am daxy" and "you are good", the second lesson includes the sentences with 4 words, etc. Note that each English sentence in this dataset includes no more than 9 words.

**Trace search for training.**    When searching for non-degenerate execution traces, the length limit of the `REDUCE` argument predictor is 2 for SCAN and the context-free grammar parsing tasks, and 5 for the compositional machine translation task. No length limit is set for the `CONCAT_M` and `CONCAT_S` argument predictors. No length limit is set for the `REDUCE` argument predictor during the inference time, which allows it to produce degenerate execution traces.

For each training sample, the model searches for at most 256 execution steps to find a non-degenerate trace, and if no such trace is found, a degenerate trace is used for training. The execution steps are counted by the number of operators, e.g., the trace in Figure 5 includes 21 execution steps. We use a simple trick to further speed up the trace search. Note that when the sequence generated in an intermediate execution step is already not a substring of the ground truth output sequence, this operation cannot be correct. In this case, we backtrack to the previous step, and sample another different operation to execute.

**Other training hyper-parameters.**    We train the model with the Adam optimizer, the learning rate is 1e-3 without decaying, and the batch size is 256. We do not use dropout for training. The model parameters are uniformly randomly initialized within [-1.0, 1.0]. The norm for gradient clipping is 5.0. We perform an evaluation for the model after every 200 training steps, and the model usually converges to the optimum within 3000 training steps.

**Model hyper-parameters.**    Each bi-directional LSTM used in the neural controller includes 1 layer, with the hidden size of 256. The embedding size is 512.

## F    More Results on the Context-free Grammar Parsing Task

In Table 7, we present the results including all different setups and baselines in [8]. Specifically, Stack LSTM, Queue LSTM, and DeQue LSTM are designed in [18], where they augment an LSTM with a differentiable data structure.

## G    More Details of the Few-shot Learning Task

Figure 9 shows the full dataset used for the few-shot learning task in our evaluation.

Table 7: The full experimental results on context-free grammar parsing benchmarks proposed in [8].

### While-Lang

| Train | Test | NeSS (ours) | Neural Parser | Seq2seq | Seq2tree | Stack LSTM | Queue LSTM | DeQue LSTM |
|---|---|---|---|---|---|---|---|---|
| Curriculum | Training | **100%** | 100% | 81.29% | 100% | 100% | 100% | 100% |
| | Test-10 | **100%** | 100% | 0% | 0.8% | 0% | 0% | 0% |
| | Test-100 | **100%** | 100% | 0% | 0% | 0% | 0% | 0% |
| | Test-1000 | **100%** | 100% | 0% | 0% | 0% | 0% | 0% |
| | Test-5000 | **100%** | 100% | 0% | 0% | 0% | 0% | 0% |
| Std-10 | Training | **100%** | 100% | 94.67% | 100% | 81.01% | 72.98% | 82.59% |
| | Test-10 | **100%** | 100% | 20.9% | 88.7% | 2.2% | 0.7% | 2.8% |
| | Test-100 | **100%** | 100% | 0% | 0% | 0% | 0% | 0% |
| | Test-1000 | **100%** | 100% | 0% | 0% | 0% | 0% | 0% |
| Std-50 | Training | **100%** | 100% | 87.03% | 100% | 0% | 0% | 0% |
| | Test-50 | **100%** | 100% | 86.6% | 99.6% | 0% | 0% | 0% |
| | Test-500 | **100%** | 100% | 0% | 0% | 0% | 0% | 0% |
| | Test-5000 | **100%** | 100% | 0% | 0% | 0% | 0% | 0% |

### Lambda-Lang

| Train | Test | NeSS (ours) | Neural Parser | Seq2seq | Seq2tree | Stack LSTM | Queue LSTM | DeQue LSTM |
|---|---|---|---|---|---|---|---|---|
| Curriculum | Training | **100%** | 100% | 96.47% | 100% | 100% | 100% | 100% |
| | Test-10 | **100%** | 100% | 0% | 0% | 0% | 0% | 0% |
| | Test-100 | **100%** | 100% | 0% | 0% | 0% | 0% | 0% |
| | Test-1000 | **100%** | 100% | 0% | 0% | 0% | 0% | 0% |
| | Test-5000 | **100%** | 100% | 0% | 0% | 0% | 0% | 0% |
| Std-10 | Training | **100%** | 100% | 93.53% | 100% | 0% | 95.93% | 2.23% |
| | Test-10 | **100%** | 100% | 86.7% | 99.6% | 0% | 6.5% | 0.1% |
| | Test-100 | **100%** | 100% | 0% | 0% | 0% | 0% | 0% |
| | Test-1000 | **100%** | 100% | 0% | 0% | 0% | 0% | 0% |
| Std-50 | Training | **100%** | 100% | 66.65% | 89.65% | 0% | 0% | 0% |
| | Test-50 | **100%** | 100% | 66.6% | 88.1% | 0% | 0% | 0% |
| | Test-500 | **100%** | 100% | 0% | 0% | 0% | 0% | 0% |
| | Test-5000 | **100%** | 100% | 0% | 0% | 0% | 0% | 0% |

Figure 9: The full dataset used for the few-shot learning of compositional instructions. This figure is taken from [28], where the percentage after each test sample is the proportion of human participants who predict the correct output.

.