[Reviews · NeurIPS 2020]

Review 1

Summary and Contributions: The paper proposes a new neural-symbolic approach to achieving compositional generalization, in particular in sequence to sequence tasks. The proposed method (NeSS) is evaluated on 1) SCAN (a popular dataset in compositional generalization research) 2) English-French machine translation task which requires compositional generalization (proposed by Lake and Baroni 2018) and 3) context-free grammar parsing tasks (Chen et al, 2018) introduced in the context of program induction. In all of the reported experimental results NeSS achieves 100% accuracy. The contributions of the paper are as follows: 1) New insights into the biases that lead to compositional generalization in neural (and neural-symbolic) methods. The authors formalize the idea of operational equivalence and show that it is important for length generalization. The idea builds upon local equivariance proposed by Gordon et al.,, 2020. The authors show that local equivariance is insufficient to achieve length generalization in sequence to sequence tasks. 2) The neural-symbolic method NeSS which incorporates the concept of operational equivalence. The method is clearly explained in the submission. The authors show that it retains the expressive power of existing sequence-to-sequence models with improved generalization. 3) Experiments using three tasks and positioning their work in comparison to the existing methods. In SCAN, the proposed method achieves the same accuracy as Synth with search across all tasks (Nye et al, 2020) but NeSS is argued to be more versatile in terms of the tasks it can be applied to and it does not require a search procedure. Similarly, in the context-free grammar parsing tasks NeSS perfoms on par with Neural Parser (Chen et al, 2018) while it is argued to support more applications than the Neural Parser model.

Strengths: - Research on neural-symbolic approaches (in the context of program induction and sequence to sequence tasks) and insights into inductive biases for compositionality are of high interest to the community. - Building connections to the work in classical NLP (Combinatory Categorial Grammar, unsupervised parsing, grammar induction) and possibly compiler design. - The paper is very well-written and there is sufficient information to understand the contributions in the main submission.

Weaknesses: - The experimental results lack error bars. - A more thorough comparison with the existing neural-symbolic methods in VQA and their biases for compositional generalization (Neural-symbolic concept learner by Mao et al 2019, Neural-symbolic VQA by Yi et al 2018, MAC by Hudson et al 2018) would strengthen the paper.

Correctness: The claims and empirical methodology (apart from the lack of error bars) seem correct.

Clarity: The paper is very well-written and clearly structured.

Relation to Prior Work: There is a discussion of related work. In the context of the methods which perform on par with NeSS (Synth with search and Neural Parser), it would be very useful to discuss the tradeoffs and applications, for instance in which circumstances the need for a search procedure is problematic.

Reproducibility: No

Additional Feedback: It would be interesting to evaluate NeSS on few-shot learning of compositional instructions (Lake et al, 2019).


Review 2

Summary and Contributions: This paper introduces a neuro-symbolic stack architecture for compositional generalization. The model solves seq2seq problems through a sequence of operations that manipulate a stack and control input/output flow. Many of these operations are themselves learnable neural modules. The paper presents impressive results on SCAN, a small-scale MT task, and a grammar learning task. This is an impressive paper. The proposed architecture is novel and introduces a number of interesting ideas for tackling compositional generalization. The paper is well-written and clear. I would like to see this work published at NeurIPS.

Strengths: I see a number of strengths to the architecture: - seq2seq models are a special case of NeSS, and thus this architecture is more general than many others applied to SCAN - The operators provide an interesting inductive bias. Some are symbolic stack operations, yet most of the operators are not hard-coded. Instead they are learnable neural modules (CONCAT, REDUCE, etc.). - The latent category predictors are a clever idea. - Search at training time alleviates the need for strong supervision regarding the execution trace

Weaknesses: A weakness of the paper is that it doesn't currently illuminate *why* your model works. The architecture is complex, and no ablation studies are provided. Are the category predictors needed? Each of the operators? Is the curriculum learning needed? What if the encoder sees more information than the limited view of the internal states/input that it currently receives? I hope some of these experiments could be provided in revisions. Relatedly, I am not sure why the model is successful on length generalization, a task where other approaches have failed. The paper mentions that the categorization of "twice" and "thrice" together is critical--is this the only reason? Can the architecture generalize to longer sequences in other cases? Does it fail without the category predictors?

Correctness: Yes

Clarity: Yes

Relation to Prior Work: Yes

Reproducibility: Yes

Additional Feedback: Although there is room for improvement, this is a strong paper that would make a nice contribution to NeurIPS. Minor comments: - pg. 1 "typically lack such generalization ability" -> "typically lack such generalization abilities" - pg. 1 "we design a curriculum learning scheme, which resembles the concept learning process for humans." I would remove/revise this statement, as it's not supported in the cognitive science literature. The "concept learning process" is arguably all of cognitive development, and your curriculum learning scheme certainly isn't capturing all of cognitive development. - Figure 1: should "Step 2: REDUCE[JUMP]" instead be "Step 2: REDUCE[jump]", since REDUCE is a function applied to the input language (lower case) to produce the output language (caps)? - Table 3: "Meta seq2seq" shouldn't be included in the upper half of Table 3, since it's not defined without data augmentation. It's simply equivalent to seq2seq without data augmentation. - pg. 7: a few more words about how accuracy is measured in experiment 4.2 would be useful * After rebuttal * Thanks for your response. The authors ran some additional ablation experiments, and promised to add this to the paper, which will strengthen it. I already thought it was a good paper and will keep my score.


Review 3

Summary and Contributions: This paper proposes a model for translation or parsing tasks specifically designed to provide compositional generalization. The proposed model, the Neural-Symbolic Stack Machine (NeSS), consists of a neural network which controls the execution of a symbolic stack machine that can manipulate sequences. The model is trained without explicit supervision on execution traces. After training, the proposed approach achieves perfect generalization on its three test domains. Results seem to be due to two features: the symbolic stack machine, which supports general recursion and the types of sequence manipulation operations required for the target domains, and a notion the authors call “operational equivalence,” in which execution traces with common features can be used to group words together in categories. These categories are used to predict source and target token categories, which help with inference.

Strengths: The results seem strong; the model generalizes well across the test domains. The fact that the REDUCE operation can implement any sequence to sequence transformation is nice -- it means that this method can take advantage of improvements in neural seq2seq performance as these methods scale, while maintaining the recursive and programmatic aspects of the stack machine approach. Significance: The problem of compositional generalization has seen much attention recently, and this work provides a neuro-symbolic framework to tackle this problem.

Weaknesses: I think an ablation without learning the syntactic categories is very important. The paper claims that operational equivalence “is key to enable generalization beyond primitive substitutions,” however this claim should be tested with a clear ablation study. I suspect that the trace search is an important determiner of the final performance of the method, and details of its implementation are not entirely clear. For instance, what does it mean to “prioritize” the usage of the extracted rules in the search? Without code or a more detailed description, it would not be possible to reproduce this search from the description provided. All of the evaluated domains are synthetic. Are there naturalistic domains for which this approach could be useful? Showing the utility of this approach for real-world problems would greatly increase the strength of the paper.

Correctness: The paper’s methodology seems fair and correct.

Clarity: The paper is relatively clear. I especially thought Figure 1 was a very good way to explain the algorithm and show how it worked.

Relation to Prior Work: I think the paper does a good job of explaining the differences between this work and prior work.

Reproducibility: No

Additional Feedback: I think it’s inappropriate to make the claim that “NeSS does not make assumptions about the grammar format.” As stated, “operational equivalence” is a necessary component of the approach, therefore the types of operational equivalence supported affect the types of generalizations possible. What are the rules and syntactic categories extracted for other domains? It would be interesting to see if these always line up with our intuitions, as they do in the SCAN domain. Overall, I think this is interesting work which could additionally benefit from ablation studies and more clarity on the train-time search procedure. --------- After author response: Thank you for your response. The authors ran an ablation study without category prediction, and promised to open-source the code in order to make the trace search procedure more transparent. I think both of these additions will strengthen the paper.


Review 4

Summary and Contributions: In this work the authors build on efforts to improve generalization in transduction-style tasks, first by incorporating a stack machine into existing seq2seq transduction models (this allows for recursion and other compositional generalization), and secondly by explicitly formalizing the notion of operational equivalence, and categorizing elements/phrases based on their compositoinal similarities. These are well-chosen additions for dealing with SCAN-like transduction problems, and the authors show significant improvements here and on two other synthetic tasks. The authors also use and discuss the importance of curriculum learning, though it would be difficult to really count this as a contribution given its usefulness in other stack/program induction work.

Strengths: The particular focus of this work, on understanding equivalent operations and incorporating this into learned computational routines, is an important problem and could be used in a variety of subfields. It's undeniably relevant to the NeurIPS community. The particular idea of learning/utilizing equivalence and its importance to generalization is sound and well-motivated, and I think there is conceptual novelty there. This is also shown to be empirically useful on the SCAN task, among other synthetic tasks, which is an established benchmark with a good amount of prior work.

Weaknesses: If one approaches this paper from the line of research of Lake and others, it's a significant improvement over existing work which aims to generalize better on the SCAN data, sometimes from a very small number of important examples in the training data. In this context, the introduction of a stack and explicit modeling of equivalence is a clear contribution and a pretty solid paper. What I'm more concerned with is approaching this paper from the other line of research, which has been trying to augment neural models with stacks or other differentiable datastructures to enable better test-time generalization, sometimes to the same goals pursued here -- like longer sequence lengths, and recursion. To this end the paper doesn't do an adequate job of discussing similar models or comparing against them in the experiments section. It seems approach taken here could have advantages over these, though they seem quite similar. I discuss this a bit later in the questions to authors, as it could change my opinion on the paper to understand where it fits into this larger set of research. EDIT: The authors clarified in rebuttal that the supplementary material models were in fact the unbounded transduction models and I'm happy to see those results and also a response from the authors on the natural followup question -- how well those baselines perform on the other tasks. This is a surprising result to me, and in essence has turned a weakness into a strength, and raises my opinion of the proposed method considerably. I suppose a natural follow-up question based on the literature in this area would be: is it fair to compare the proposed method to the baseline stack LSTM without a multi-pop type of operation or multi-word decoding. However I don't feel any aspect of whether to accept this paper would hinge on this aspect of the experiment design.

Correctness: It seems generally correct.

Clarity: Mostly. The part I found most lacking is that throughout reading the paper I'm aware that there's some sequential model with some stack augmentation, but there's no discussion of the actual modeling components until rather late (bottom of pg 4) and really no quick overview of the architecture early on. I think it's burden on the reader to get through the intro and even into talk of equivalence without understanding more about what the model looks like. I also didn't get much out of Figure 2 as is, and I'm familiar with the CCG-style notation. I feel like the relationship between Cs_1 and argument categories wouldn't be clear here, or that this figure wasn't tied into the exposition as well as it could have been. I would think the mix of CCG and more proprietary style of presenting info in Fig 3 or 4 is just all around doesn't help to build up a consistent story. These all could have been a bit more cohesive IMO.

Relation to Prior Work: Related work is mentioned in passing, but only previous work on SCAN is discussed in-depth and compared against.

Reproducibility: Yes

Additional Feedback: My most pressing question is also how I would like to see the related work section expanded -- the authors chose not to compare against other neural sequence models that operate with stacks, either explicitly with a differentiable stack (work cited in the paper) or those that have some structural biases for more stack-like behavior (ordered neurons, etc.). Is there a reason these systems couldn't be use for these transduction tasks and their numbers reported? If there is not a substantial hurdle, I would insist that such systems be included in the comparison. It is very difficult to argue for accepting the paper without them. If there are difficulties in doing so that aren't clear, I think these should be made apparent. Such comparisons would also make a stronger case for the empirical importance of more explicit equivalence and categorization handling within the model, though it seems that there would also be ways of putting more of these into existing work (through the inclusion of VQ-VAE type components, though differentiable stack models seem difficult enough to optimize as is). I realize these methods don't do trace search, which would put them at an advantage -- but is also a testament to their generalization, even over the method presented here. The other thing I'm a bit concern about is these single instance of important concepts in the SCAN dataset. If making the above comparison, I would just want to be sure that such data is perhaps over-sampled when compared against systems that use a bit more grammar/hand-holding during training that might accomplish a similar goal of paying more attention to those examples. I also feel like the numbers bear out the success of the method, but we're not left with a clear understanding of why -- what the model inferred to be categorically similar, and where that resulted in an impovment over the baselines. Is table 2 the best we can get? I was under the impression SCAN was more difficult than this, or similarly that this wouldn't be a significant enough advantage to warrant such large quantitative improvements. Smaller comments: L48: a key component to achieve_ / to achieving? L70: a lot of this could be clearer. It seemed like Op should be parameterized by pi. What is the mapping on L_s -> L_t, over words? 1-to-1? L136: minus "the" L262: for SCAN domain L289: without the supervision on

[Author Response · NeurIPS 2020]

We thank all reviewers for constructive comments. We first address the common issue, then the individual questions.

**Common issue: the importance of category prediction.** Category prediction is required to determine the order to
apply different parsing rules. We added an ablation study on the SCAN length split to demonstrate its importance. With
category predictors, NeSS achieves 100% test accuracy in 5 independent runs. Without category predictors, NeSS still
achieves 100% test accuracy in 2 runs; however, the accuracy is around 20% in the other 3 runs. The main reason is that
without category predictors, the model may not learn that the parsing rule for "thrice" has the same priority as "twice".
For example, in the test set, there is a new pattern "jump around right thrice" that does not appear in the training set. The
correct translation is to parse "jump around right" first, then repeat the action sequence thrice, resulting in 24 actions.
Without category prediction, a model could mistakenly parse "right thrice" first, concatenate the action sequences of
"jump" and "right thrice", then repeat it for four times, resulting in 16 actions. This model could still achieve 100%
training accuracy, because this pattern does not occur in the training set, but the test accuracy drops dramatically due to
the wrong order for applying rules.

However, **category prediction alone doesn't guarantee length generalization.** A model can still fail by parsing rules
incorrectly on longer test samples, e.g., they may predict "X Y" as the action sequence when the input is "X after Y".
Recursion and sequence manipulation supported by NeSS are critical to learn such parsing rules to generalize. Based
on the reviewers' suggestions, we will add more discussion and ablation studies.

**R1. [Q: Few-shot learning and error bars?]** We evaluated NeSS on the few-shot learning task in (Lake et al, 2019),
and it again achieved 100% accuracy. Regarding error bars: for SCAN and context-free grammar parsing, the test
accuracy of NeSS is 100% in 5 independent runs; for compositional machine translation, the exact match accuracy of
NeSS is 100% in 2 runs, and 62.5% in 3 runs. As discussed in [29], only one reference translation is provided for each
test sample, but there are 2 different French translations of "you are" that appear frequently in the training set, which are
both valid translations. When the model predicts the alternative translation, the exact match accuracy becomes lower.

**[Q: More discussion of related work?]** Neural-symbolic methods for VQA consider the generalization to new
composition of visual concepts, and scenes with more objects than training images. Compared to VQA, our tasks do
not require visual understanding, but need much longer execution traces. Compared to Synth, which searches for >1000
programs on SCAN, models that do not need an extensive search during inference could be more efficient, especially
when the task requires more examples for test-time input specification. We will add more discussion in our revision.

**R2. [Q: All Components needed?]** See the common response above for the importance of category prediction, and
we will incorporate other writing suggestions in our revision. All operators are required for sequence manipulation
using the stack machine. Curriculum learning is required for the model to find correct traces for long training inputs.
When the encoder receives more information, e.g., the entire input sequence, the model does not utilize the recursion
property of the stack machine anymore, and thus the generalization accuracy becomes similar to a seq2seq model.

**R3. [Q: Experiments on naturalistic domains?]** Our goal is to address the compositional generalization problem.
However, most existing natural language benchmarks are not designed for this purpose. The challenge in those datasets
is to handle the inherently ambiguous and potentially noisy natural language inputs. Their training and test sets are
usually from the same distribution, and thus do not evaluate compositional generalization. Therefore, we did not
run experiments on these datasets. Instead, our evaluation focuses on the standard sequence-to-sequence generation
benchmarks used in previous works on compositional generalization. Such benchmarks are typically constructed with
synthetic grammars, so that it is easier to change training and test distributions. We consider improving compositional
generalization for more natural inputs as future work. We will clarify these points in our revision.

**[Q: Trace search details?]** When the current machine status is included in the rule set extracted from previous lessons,
NeSS directly applies the rule, and only searches for other operations when it cannot find any consistent trace. We will
discuss more details about the trace search and open-source the code in the final version.

**R4. [Q: Comparison with differentiable data structures?]** In Appendix E, we quoted the results of differentiable
data structures on context-free grammar parsing from [9], denoted as Stack LSTM, Queue LSTM and DeQue LSTM.
These results show that a stack alone is insufficient to obtain good results. We also evaluated these models on other
benchmarks, and the results are similar to seq2seq. For example, stack LSTM achieves 100%/17%/0%/0.3% test
accuracy on Simple/Length/Jump/Around Right splits of SCAN, though the training accuracies are always 100%.
These results showed that without enhancing the stack machine with more operators for sequence manipulation, simply
augmenting neural networks with a stack alone is not enough to achieve good generalization on these tasks. We will
provide a more detailed discussion of the related works and the new results in our revision.

**[Q: Data is over-sampled?]** See common response for the discussion of accuracy improvement. Compared against
prior works, we used the original training set, and we didn't do any additional sampling during training. The mapping
from $L_s$ to $L_t$ is over sequences, and we will incorporate other writing suggestions in our revision.

[Meta-Review · NeurIPS 2020]

The paper proposes a new method for compositional generalization in sequence-to-sequence tasks. The basic idea is to have a symbolic stack machine (capable of compositionally manipulating sequences) that is controlled by a neural network. The method gets perfect accuracy on an existing compositional generalization dataset, a small-scale English-French machine translation task, and a grammar parsing task. The paper was well-received. Pros: + Novel architecture + Attractive way of providing inductive bias without hardcoding too much knowledge + The paper is well-written + Strong experimental results in the domains considered Cons: + The paper could do more by the way of providing insights about why the model works. The reviewers appreciated the clarifications provided in the author feedback. Please integrate these clarifications (in particular, the results of the ablation study and comparisons with differentiable data structures) into the main paper, and consult the reviews for more detailed feedback.